# Gait Characteristics Analyzed with Smartphone IMU Sensors in Subjects with Parkinsonism under the Conditions of “Dry” Immersion

**DOI:** 10.3390/s22207915

**Published:** 2022-10-18

**Authors:** Alexander Y. Meigal, Liudmila I. Gerasimova-Meigal, Sergey A. Reginya, Alexey V. Soloviev, Alex P. Moschevikin

**Affiliations:** 1Medical Institute, Petrozavodsk State University, 33, Lenina pr., 185910 Petrozavodsk, Russia; 2Physical-Technical Institute, Petrozavodsk State University, 33, Lenina pr., 185910 Petrozavodsk, Russia

**Keywords:** inertial measurement unit, smartphone, accelerometry, TUG test, gait, Parkinson’s disease, “dry” immersion

## Abstract

Parkinson’s disease (PD) is increasingly being studied using science-intensive methods due to economic, medical, rehabilitation and social reasons. Wearable sensors and Internet of Things-enabled technologies look promising for monitoring motor activity and gait in PD patients. In this study, we sought to evaluate gait characteristics by analyzing the accelerometer signal received from a smartphone attached to the head during an extended TUG test, before and after single and repeated sessions of terrestrial microgravity modeled with the condition of “dry” immersion (DI) in five subjects with PD. The accelerometer signal from IMU during walking phases of the TUG test allowed for the recognition and characterization of up to 35 steps. In some patients with PD, unusually long steps have been identified, which could potentially have diagnostic value. It was found that after one DI session, stepping did not change, though in one subject it significantly improved (cadence, heel strike and step length). After a course of DI sessions, some characteristics of the TUG test improved significantly. In conclusion, the use of accelerometer signals received from a smartphone IMU looks promising for the creation of an IoT-enabled system to monitor gait in subjects with PD.

## 1. Introduction

Parkinson’s disease (PD) is very suitable for the application science-intensive instrumental research methods. PD is gradually becoming a kind of “model disease” for the testing of new technologies for PD diagnostics and escorting PD subjects [1]. For several reasons, PD is one of the most studied neural pathologies in humans. One of the reasons is that PD is a widespread neurodegenerative disease worldwide, and its prevalence is increasing [2]. Furthermore, PD exerts a high economic burden on society [3] and worsens the quality of life of patients with PD [4]. Next, PD is characterized by gradual progression over decades [5], and PD symptoms are reliably quantified using clinical scales, which allow for the mathematical modeling of PD evolvement [6]. In addition, PD seems to be an extremely informative research object, since it allows for the development of insights into such phenomena as muscle tone and tremor, motor commands, postural reactions, orientation in space and gait.

Earlier, we have shown that in subjects with PD that both single session of Earth-based microgravity—modeled with “dry” immersion conditions (DI) [7]—and a program of repeated DI sessions [8] attenuates muscle rigidity and tremors and improves some aspects of activity of daily living. Additionally, some motor-cognition tests [9] and characteristics of hemodynamics and heart rate variability were improved after a program of DI sessions [10]. On the other hand, the function of a patient’s spatial orientation in a vertical stance and the function of postural transition proved non-responsive to the condition of DI [7]. Muscle rigidity is often associated with bradikinesia (slowness of movements) and akinesia (difficulties with starting motion), which is seen in the akinetic-rigid form of PD. This allows for the presumption that a decrease in muscle rigidity, provoked by DI conditions, may result in an improvement of gait characteristics, e.g., gait speed, cadence and length of steps.

The Timed Up-and-Go (TUG) test has proven to be reliable in many domains of neuromuscular and orthopedic pathology for assessing gait, basic mobility skill, strength, agility and balance [11]. It consists of five sequential phases: (1) standing up from a chair (Sit-to-Stand transition phase), (2) walking straight forward (Gait-Go phase, including stand-to-walk transition), (3) turning by 180° (U-turn phase), (4) walking back (Gait-Come phase), and (5) sitting down (Stand-to-Sit transition with a turn). In its classic 3 m form, the TUG test provides an immediate score, requires no training and only needs one tester [11]; however, the classic TUG test supplies little data on gait as it requires the patient to take only 4 to 6 steps in both directions. In addition, the first step, the step prior and right after the U-turn, and the last step are clearly specific by their biomechanical functionality (transition to locomotion, decelerating when approaching the U-turn point and the end-point of the test, correspondingly). To overcome this problem, longer (expanded) versions of the TUG test were invented. For example, Haas et al. [12] presented the so-called L-test, which includes longer walks and turning in both directions, and Galán-Mercant et al. [13] presented a 10 m version of the TUG test. Earlier, we proposed an even longer (extended) version of the TUG test (13 m long, which returns around 20 steps in one direction) to provide a more precise view of a self-paced walk at a comfortable speed in the middle of both the Gait-Go and Gait-Come phases [14].

Throughout the last decade, instrumented versions of the TUG test (iTUG) were increasingly invented. In most of these versions, varied numbers and positions of miscellaneous inertial measurement unit (IMU)-based wearable sensors (accelerometers) were used to discriminate between the phases of the TUG test [15,16]. The IMU is often fixed on a foot to obtain the exact position of a limb in real time. The quality of the sensor’s trajectory restoration is often controlled by video capture, and its position accuracy is in the millimeter range [17,18]. One of the problems with such a system is the time synchronization of inertial sensor data and video flow [19]; some researchers use multi-IMU networks. For example, in the study by Qiu et al. [20], a system of 100 Hz IMUs connected via WiFi was applied for monitoring complex gait parameters, including knee angle dynamics. Bogaarts et al. [21] explored the impact of noise on gait features that had been extracted from smartphone sensor data. They created a model of a moving body, generated acceleration signals for plenty of the points on the body, then added noise to simulated signals and after that tried to extract the gait features. As a result, they showed that sensors in from-the-shelf smartphones are sufficient for registering acceleration signals, given that the sensor’s noises introduce negligible impacts on the computation of step power and other similar parameters [21].

There is a multitude of technological approaches for studying PD. Among them are optical motion trackers, biopotential devices, audio and video recording, and, especially, wearable sensors, such as smart glasses, hats, insole sensors of ground-reaction force and smartphones [22]. Previously, we evaluated the effect of DI on PD subjects with conventional laboratory tools (EMG, reaction time, tapping test, posturography) [7,8,9,10].

According to the review by Deb et al. [22], wearable sensors are currently the most used (40% articles in the field), while smartphones are the least used; however, starting from the year 2020, there is a trend in the growing number of articles that have used smartphones for their research [22]. Among the application areas, the diagnosis and monitoring/prognosis of PD were the most studied, and among symptoms, the gait, tremor and speech of a subject were the most studied [22]. Smartphone applications have good to excellent ability for predicting and discriminating gait and postural instability between PD subjects and healthy controls, as well as the leg dexterity and gait cycle breakdown between PD subjects with different severities of the disease [23]. Thus, there is strong evidence regarding the potential use of smartphone applications to assess gait and balance among individuals with PD in the home or laboratory [22,23].

Smartphones are equipped with IMUs that consist of a 3-axis accelerometer, a 3-axis gyroscope and a digital magnetometer that is comparable in sensitivity to research-grade biomechanical instrumentation [24]. In the study by Manor et al. [24], smartphones were placed in the front pocket, which is relevant for non-laboratory settings. Typically, smartphones are secured to the trunk or lower extremities. In our earlier study, we suggested a method of reconstruction of the head trajectory in 3D-space using the IMU-based accelerometer and gyroscope of a smartphone, which was fixed on a subject’s head [14]. We assumed that, in accordance with the concept of the “inverted pendulum model”, the head produces the biggest displacement in the vertical axis [25]. Similarly, Hwang et al. [26] conducted research with a 60 Hz single IMU fixed on a head, which is similar to the method used in our study. They used a FIR low-pass filter to reduce the noise and applied threshold to capture the exact phases of a stride. Since a FIR filter introduces a certain latency in the processed series data, this fact should be considered in data analysis. The authors also presented comprehensible figures that demonstrated that a single sensor fixed on a head picks up acceleration signals from both legs, and further, they clarified how this obtained signal might be “decoded” and understood. Thus, a smartphone fixed on a subject’s head can return meaningful information about gait. Still, the 100 Hz IMU of a smartphone does not allow for sufficiently precise tracking of the trajectory of the head.

The major hypothesis of the present study was that gait characteristics in patients with PD are responsive to the conditions of either one session of DI or a course of repeated DI sessions. To address this, we obtained up-sampled acceleration signals from smartphone-based 100 Hz IMU sensors attached to the subject’s head during a 13 m TUG test before and after single session of DI and a program of seven DI sessions.

## 2. Materials and Methods

### 2.1. Subjects

Altogether, data from six PD subjects was collected in the study. Six subjects with PD participated in the study after providing their informed consent. Their anthropometric and clinical data and the medication they use is presented in Table 1. All of them are from the same cohort of subjects who participated in our earlier studies [7,8,9,10]. The data on gait characteristics presented in this article were obtained from these studies. All subjects signed their informed consent, and the protocol of the study was approved by the Local Ethical Committee (joint ethics committee of the Ministry of Healthcare of the Republic of Karelia and Petrozavodsk State University (Statement of approval No. 31, 18 December 2014)).

### 2.2. Procedures

#### 2.2.1. On-Earth Model of Microgravity

The on-Earth microgravity was modeled using the conditions of a “dry” immersion (DI). This method of DI has already been presented in detail in our earlier papers [7,8,9,10]. In brief, the condition of DI was created with the help of MEDSIM (Medical simulator of weightlessness, Center for Aerospace Medicine and Technology, IMBP, Moscow, Russia), which is housed in the Laboratory of Novel Methods in Physiology (Petrozavodsk State University). The MEDSIM facility uses a bathtub filled with 2 m^3^ of fresh, thermally comfortable water stabilized at T = 32 °C. The water in the tub was periodically filtrated and aerated to prevent bacterial contamination. The water surface was covered with a large, square waterproof film (3 × 4 m^2^), which was wrapped around the subject’s body. The DI session was conducted at 9:30 AM, in the condition of “on-medication” in order to synchronize the effects of DI and the anti-PD therapy. The subjects usually took their medicines 2 h before the study, at 7:30 AM. Before DI, subjects were instructed to drink 200 mL of water and urinate due to the strong diuretic effect of DI [27]. Before immersion, subjects laid supine for approximately 10 min on a solid movable motor-driven platform on a cotton sheet in the MEDSIM facility in order to attach electrocardiogram electrodes, measure brachial blood pressure (BP), and familiarize (altogether around 5 min) and note ECG recordings in standard lead II (5 min). If after 10 min of lying supine the subject’s BP was higher than 140/80 mm Hg, he/she was not allowed to enter DI and the study was postponed for another day. After that, the platform was driven to its bottom position, and subjects found themselves immersed in water without direct contact with the water; the head and upper chest were left above the water’s surface. One DI session lasts for 45 min. BP and ECG were monitored at the 15th, 30th and 45th min. After the DI session, subjects laid motionless on the platform in its upper position for a further 5–7 min for re-adaptation to the pre-DI conditions and for ECG monitoring. Altogether, 22 measurements were successfully conducted: 10 measurements before/after a single session of DI (5 paired sets of data), and 12 measurements before and after a program of DI sessions (6 paired sets of data).

The program of DI consisted of seven 45 min DI sessions that were conducted twice a week for 25–30 days. The total DI dose during the course was 5¼ h.

#### 2.2.2. Test Protocol: 13 m TUG Test

The TUG test was performed in its extended form (13 m instead of the conventional 3 m long test). Still, its phases were all the same: (1) standing up from a 46 cm highchair (Sit-to-Stand phase), (2) walk straight (13 m, Gait-Go phase, including Stand-to-Walk transition), (3) turning by 180° (U-turn), (4) walking back (13 m, Gait-Come phase), and (5) sitting down (Stand-to-Sit transition with a turn). In addition, unlike the classic 3 m version, the 13 m version of the TUG test allowed for an analysis of the subject’s steps (gait)—because subjects performed up to 20 steps in one direction, which is sufficient for analyzing gait [28]. The TUG test was performed 15 min prior and then 8–10 min after the DI session. A baseline gait analysis throughout the day was not conducted, neither was one conducted before or after the DI session. The TUG test was performed prior and 8–10 min after the DI session.

### 2.3. Data Processing

During the TUG test, the acceleration and rotation rate were measured with the sensor module in the smartphone Xiaomi Mi4 (Xiaomi Tech, Bejing, China 68.5 mm × 139.2 mm × 8.9 mm, 149 g). The obtained signals were further processed offline. 

The subject was instructed to sit still and look forward before and after the test. In a motionless state, the shape of the accelerometer signal is formed by the current projections of the gravity vector, measurement noise and the existing zero-G offsets, as well as the head tremor. The beginning and end of the movement are characterized by a change in the *x*-, *y*- and *z*- components of the acceleration vector due to the inclination as well as the presence of linear accelerations while standing up and sitting down. For the gyroscope in a motionless state, the signal includes the sensor noise as well as the head tremor; however, it is characterized by the constancy of the mean value (measurement offset). The start and end points of the TUG test were selected manually by analyzing the change in the mean value due to the rotation of the head and body during inclination while standing up and sitting down.

The internal phases of the TUG test and step moments were determined automatically. For steps in a straight-line walk (15–17 more or less uniform steps in the middle of both the Gait-Go and Gait-Come phases), a set of gait features was calculated. Altogether, subjects performed 35–40 steps in both directions, of which 30–35 steps that were in the middle of the walk were analyzed. 

### 2.4. Inertial Data Acquisition and Pre-Processing

The sampling rate of the inertial sensors—both the accelerometer and gyroscope—of the Xiaomi Mi4 smartphone that was used in the present study was 100 Hz (the period between data samples was Δt = 10 ms), which can be regarded as neither reliably accurate nor fast. The smartphone was fixed on the back of the head of a subject with an elastic band and, additionally, a tight-knitted hat; subjects felt comfortable with this kind of fixation and the smartphone never fell out of its position. Values for the acceleration and angular velocity were collected as a time-stamped data stream. Thus, the measurements were accompanied by timestamps from the smartphone’s operating system timer. For further analysis, the accelerometer and gyroscope measurements with a time-stamp difference of less than 5 ms (half of the measurement period) were considered synchronous. In order to increase the time resolution and achieve a smoother distribution, the time series data were up-sampled to a 10-fold-higher frequency of 1 kHz (Δt’ = 1 ms) (Figure 1). In addition, an increase in the time resolution allowed for the application of high-order digital filters to the obtained time series.

Since the analyzed signal tended to be periodic, the up-sampling, which used Fast Fourier Transform, could be applied. Furthermore, as long as the measurement signals are real-valued, the real (single-sided) FFT is suitable for conversion into the frequency domain. In the frequency domain, up-sampling means there is zero-padding at the end of the high frequency components of the signal. The up-sampling procedure included the following steps:(1)forward Fourier transform of original signal X = F(*x*);(2)zero-padding F(*x*) up to new length F(*y*);(3)inverse Fourier transform of F(*y*) to obtain up-sampled signal *y*;(4)scaling up-sampled signal *y* to preserve amplitudes.

A simple calibration of the zero offsets of the sensors was performed before the start of the test. To do this, we used the measurements obtained from a smartphone placed on a horizontal surface. It was noted that the sensor offsets were probably pre-calibrated by the Android OS. The bias instability and velocity/angle random walk for smartphone sensors was previously analyzed by us using the Allan variation [14]. The bias instabilities are (7.3, 8.2, 8) × 10^−4^ m/s^2^ for the *x*-, *y*- and *z*-axis of the accelerometer, and (1.7, 5, 7) × 10^−5^ deg/s for the gyroscope. Since the test duration is less than 1 min, the bias drifts can be neglected.

The orientation of the smartphone was calculated using a well-known complementary filter proposed by Robert Mahony et al. [29] and is expressed in the form of a quaternion Q. Using Q, the acceleration and angular velocity measurement vectors were converted to a global coordinate system (global frame):GV = Q ⨂ SV ⨂ Q*,(1)
where GV and SV are “pure” quaternions associated to the 3-dimensional measurement vector in the sensor frame and global frame, respectively; Q* is a conjugate of Q; and the ⨂ symbol represents the Hamilton product.

### 2.5. Turns (Rotation) Detection

Automatic detection of a turn was conducted by analyzing the projection of the angular velocity on the vertical axis. No additional filtering of measurements was performed. If the values of the amplitude and duration of the rotation rate exceeded certain threshold values, a rotation was considered to be detected (recorded). At the first stage, a comparison was made with the threshold value of the rotation rate (10 degrees per second). At the second stage, the rotation duration was estimated. Rotations lasting less than 1 s were discarded. If three or more turns were detected in the TUG test, the two longest turns were considered the 1st (at the U-turn phase) and 2nd (prior to sitting down on a chair) turn. According to the available experimental data, this algorithm was successful in 100% of cases for both turn events.

### 2.6. Step Detection

Step detection was automatically performed by analyzing the time series of the acceleration vector. Since the typical cadence of stepping is about two steps per second, the measurements were filtered with a forward–backward zero-phase low-pass filter (Butterworth, 10th order) with a cut-off frequency of 3 Hz, which allowed us to obtain the LPF time series data. After that, peak values of the filtered signal were detected. Moments where the acceleration magnitude reached 11 m/s^2^ were taken as the approximate time-stamp of the initial contact of the foot with the ground (T’ point).

For each step, the revised time-stamp T_step of the heel strike and the corresponding maximum acceleration along the vertical axis were determined by searching for the maximum value in the ±40 ms window near the T’ point. Not all steps taken during the TUG test were taken into account for gait analysis. The following local maxima that were obtained during the step detection procedure were discarded:the first peak corresponding to the moment of standing up and the second peak corresponding to the moment of the first step;steps during the first U-turn;steps from the moment of the second turn until the end of the whole test.

### 2.7. Gait Features

#### 2.7.1. Duration of the TUG Test Phases (D-Parameters)

To estimate the duration of the entire TUG20 test and its phases, the following parameters were determined (Figure 2):

D1 (The entire TUG test duration): the time from the very beginning of motion (the Sit-to-Stand movement) until the end of the test (sitting down on a chair).

D2 (Corresponds to the Sit-to-Stand phase plus the Stand-to-Walk period): the time from the start of the lifting to the moment of the heel strike on the second step.

D3 (U-turn phase, the 1st turn duration): the time to perform a 180° turn at the far turning point.

D4 (Walk-to-Sit phase, or the 2nd turn duration): the time from the beginning of the second turn until the end of the test.

#### 2.7.2. Characteristics of the Temporal Stability of Stepping (S-Parameters)

For the analysis of gait stability, only straight-line, uniform steps were taken into account (see Section 2.6). The following parameters were computed:

S1 (Mean_step_duration, s): the duration of the step (dt) was determined as the difference between consecutive time-stamps of successive steps (T_step_ moments). Before calculating the average value, two points with the largest deviation from the median value of step duration were discarded (red crosses, see Figure 2).

S2 (Step_duration_std, s) (see Figure 2).

To calculate the cadence mean and standard deviation, the “instantaneous walking pace” was first estimated for each step (cadence = 60/dt); then two outliers should be discarded. Usually, these outliers were characteristic of the “transitional” moments during the TUG test (at the beginning and end of the Gait-Go and Gait-Come phases, and before the U-turn). 

S3 (Cadence_mean, steps per min).

S4 (Cadence_std, steps per min).

S5 The ratio of the average deviation of the two largest outliers of the step duration to the standard deviation of the step duration without taking into account the two largest outliers (red double-sided arrow, see Figure 2). S5 reflects a tendency to take unusually long steps (LS). 

S5 was calculated according to the following algorithm:Select two outliers from “step duration” values;Calculate D_WO_ as the mean value of the step durations without outliers;For each outlier, calculate the absolute difference from D_WO_, then calculate the mean of differences (D_OUT_);Calculate D_WO_STD_ as the standard deviation of the step durations without outliers;Calculate LS = D_OUT_/D_WO_STD_.

The estimates of the probability density functions of the step duration and the acceleration upper/lower peaks were obtained using kernel density estimation (KDE). KDE was computed using the Python scipy.stats.gaussian_kde function (written by Robert Kern, 2004, Enthought, Inc., Austin, TX, USA). As P1, P2 and S2 values are related to the width of the target variables’ distributions, they are shown as the full width at half maximum. On the left panel, the red dots denote minimal acceleration when both feet were touching the floor, and the green dots denote heel strike. The open red and green circles represent these dots. Two outlier values are denoted with black crosses. On the right panel, the open black circles represent the individual step duration along the time course. The outlier values, denoted by red crosses (>0.7 s), represent unexpectedly longer steps right prior to U-turn. Furthermore, note that during the Gait-Come phase (upper group of open black circles), the length of the steps decreased roughly from 0.68 m to 0.6 m. For more information, see the text below; these data were obtained from Subject 6.

#### 2.7.3. Characteristics of the Power Stability of Stepping (P-Parameters)

To analyze the power (amplitude) characteristics of each step, the following parameters were estimated:

P1 (Heel_strike_accel_std, m/s^2^): the standard deviation of the vertical acceleration in Tstep moments. Two outliers were discarded. P1 characterizes the stability (uniformity) of the heel strike during stepping in the Gait-Go and Gait-Come phases (see Figure 2).

P2 (Swing_accel_std, m/s^2^): standard deviation of the vertical acceleration minima that corresponds to the weight transfer phase. Two outliers were discarded. P2 characterizes the stability (uniformity) of the minima values when both feet made contact with the floor during the swing phase of stepping during the Gait-Go and Gait-Come phases (see Figure 2).

P3 (Peak-to-peak_vertical_acceleration_mean, m/s^2^): the average difference between the minimum and maximum of the vertical accelerations in a series of straight steps.

All D- and P-parameters, and some of S-parameters, can be identified from Figure 2. The duration of the entire TUG test (D1) and its phases—D2, D3 and D4—shown on the left panel in Figure 2, were recognized automatically by analysis of the acceleration and rotation rate time series data. The upper peaks (green dots) correspond to heel strike moments and form a cloud of green open circles to the right. Their distribution is characterized by the P1 parameter. Similarly, P2 describes the width of distribution of the acceleration minima during the swing phase. The horizontal green and red dotted lines denote the medians for these sets. P3 is the difference between the medians. The right panel in Figure 2 describes the distribution of the performed steps over the step duration (histogram and kernel density estimation). S1 stands for the average step duration and S2 stands for the standard deviation. The black circles—from the lowest to the highest—correspond to the recognized steps from the first step to the last one in time. The two longest steps were excluded from the averaging statistics; however, they probably informed the light form of “freeze of gait”.

### 2.8. Statistical Analysis

The analysis was executed with IBM SPSS Statistics 21.0 (SPSS, IBM Company, Chicago, IL, USA). The values of D1–4, S1–5 and P1–3 were compared in the pairs of conditions “before-after a single DI session” and “before-after a course of DI sessions” with the non-parametric paired Wilcoxon *t*-test.

## 3. Results

None of the D, P and S gait characteristics responded to the conditions of a single (“acute”) DI session (Table 2); however, in each of the five examined subjects with PD, at least several—usually different—parameters were positively modified after a 45 min session of DI. Furthermore, at least in 2–4 measurements of 5, the gait parameters changed to better values. For example, in Subject 1, after a single session of DI, the D1 (duration of the entire TUG test) decreased by 5 s—from 30 to 25 s—and the subject’s cadence increased from 88 to 100 steps/min. After another DI session with this subject, the values of the P1 parameter (heel strike) increased from 0.55 to 0.77 m/s^2^, P2 increased (swing phase) from 0.50 to 0.82 m/s^2^, and P3 increased from 8–9 to 10–11 m/s^2^ (Figure 3), which provides insight into the increased variability of step length after DI and the stronger strike of the heel on the floor. In Subject 2, only the value of D4 decreased, which similarly occurred in Subject 3, wherein the value of S5 decreased (Figure 3). In addition, in all five cases of DI, the change in the distribution type of P2 from a unimodal distribution to a more bimodal one took place (see Figure 3). The individual data for all measurements are presented in Appendix A Appendix A.

Unlike with a single DI session, a course of DI sessions exerted a significant influence on a few gait parameters, namely, D4 and S5 (see, Table 2), which means that subjects with PD performed sitting down on a chair with turning (D4 phase) faster, and there were unusually long steps after a course of DI sessions. The individual plots of stepping are presented in Figure 4.

## 4. Discussion

The purpose of this study was (1) to test the reliability of an assessment of stepping characteristics with an up-sampled IMU-based accelerometer signal and gyroscope of a smartphone when placed on the subject’s head, and (2) with the help of the acceleration signal, to study the effect of a single DI session and a program of DI sessions on stepping in subjects with PD during a long version of the TUG test.

There are plenty of studies that have demonstrated sufficient reliability of iTUG test technologies based on a smartphone’s IMU to recognize the phases of the TUG test [24,30,31,32]. It has been concluded that the iTUG test is relevant for self-administered TUG test [31,33] and has good agreement with 3D motion video capture analysis [34], and it is superior to stopwatch measurement [35]. As such, the recognition of sub-phases during the instrumented TUG test, either in its classic or extended versions, with smartphone accelerometers is not necessarily novel; however, most of these studies were conducted with the classic 3 m TUG test, with a smartphone attached to a belt and with a 100 Hz sampling rate.

Instead, in the present study, we focused on (1) the gait analysis during the Gait-Go and Gait-Come phases with the help of (2) the extended 13 m TUG test, and (3) with a smartphone fixed to the head. It has been found that the 13 m TUG test returns information about 16–21 steps in each direction (28–36 steps, altogether), of which 15–17 steps in the middle of the Gait-Go and Gait-Come phases of the TUG test were considered to be functionally uniform (straight walk). This number of steps is reliable, as data collected from 10–20 strides (20 steps) were reported to be sufficient for the reliable characterization of the gait speed and cadence of stepping [28]; however, the reliable evaluation of the variability of stepping requires much more data (hundreds of strides) [28]. In addition, we considered that the pre-processing of the time series with an up-sampling procedure (from 100 up to 1000 Hz) allowed us to increase the accuracy of the capture of stepping events (heel strike and swing phase). Moreover, we paid much attention to the step variability and evolution over time. For example, we introduced a special new parameter that measures the tendency to “freeze of gait”—i.e., unusually long steps, which can provide insights into the difficulties of performing a step.

From a technical point of view, the obtained parameters and graphical presentation of the TUG test can be regarded as reliable and demonstrative, as it allows for the tracing of individual features of the subject’s gait and the recognition of graphical patterns of the subjects by eye. Furthermore, the position of the smartphone on the head can be regarded as a reliable site for the collection of information about a human’s gait. This allows for a reduction in the number of IMUs to one that is placed on the head.

We found that a single (“acute”) 45 min DI session exerted no effect on the studied parameters of gait across the entire group of subjects with PD, which is opposite to our original hypothesis. On the other hand, in each subject, an individual set of gait characteristics still changed. For example, in Subject 1 (see Figure 3), the entire duration of the TUG test (D1) decreased by 5 s (or, by 15%), and the time to perform the U-turn (D3 phase) decreased by 0.2–0.3 s, while the cadence (S1) increased by 3–12 steps/min. Furthermore, P1 (standard deviation of heel strike acceleration) and P2 (standard deviation of acceleration minima during the swing phase) increased by 0.2–0.3 m/s^2^, and P3 (vertical acceleration range) increased by 2–3 m/s^2^. As a whole, these changes suggest that after a single DI session, Subject 1 walked faster, performed faster turns and stepped more firmly on the floor. All these modifications can be regarded as positive. In Subjects 2 and 3, the effect of DI conditions was negligible, probably due to the relatively good initial values of their gait parameters, for example, in Subject 3, their cadence was 145 steps/min—in comparison to 90–112 steps/min in Subjects 1 and 2. In addition, Subject 1 did not take anti-PD medicine, which means that before the DI session he stood in the “off-medication” condition. As a result, the effect of DI was not inferred by anti-PD therapy.

The effect of a program of seven DI sessions was a bit more pronounced. At a minimum, the D4 and S5 parameters became significantly larger after a course of DI, and a change in P2 values resulted in an increase after a program of DI. The reaction to DI conditions was individually significant. Again, Subject 1 presented the most notable improvement in D1 (by 5–6 s, or by 20%), and almost in all other parameters. Subjects 2, 4 and 5 demonstrated moderate improvement of only some parameters, and Subject 3 demonstrated a notable improvement of gait.

The Internet of Things (IoT) is comprised of interconnected devices, machines, and servers with data storage that functioning through a network [36,37]. A smartphone can be considered an ideal measuring device for further instrumentation and incorporation into IoT-enabled systems since it already appears as a part of the Internet. 

A smartphone is always “at hand” (in the pocket), it is not heavy or cost-effective, and it is already pre-set for data transfer to cloud-based storage [24]. Smartphones are already efficiently used to detect and monitor PD symptoms, e.g., with reaction time tests, tapping tests, and voice (speech), posture and gait tests [38], and there are smartphone applications that are available for self-testing [39]. In a sense, smartphones have undergone a kind of “instrumentalization” compared to regular consumer devices—for example, a treadmill [40]. Altogether, this makes smartphones a relevant candidate for the implementation of diagnostic and monitoring applications in PD. Smartphones are very suitable because, unlike wearable sensors, they do not need additional resources, as they are already a part of the Internet. In addition, smartphones are capable of performing online calculations.

Data collected on the gait of PD subjects with wearable accelerometers is suitable for Artificial Intelligence (AI) or IoT decisional support [36,41]. AI-based wearable gait monitoring is already used for optimization of Parkinson’s disease management [41]. We figured that smartphone applications based on AI can be applied to monitor gait characteristics in PD subjects. Among the varied learning methods, deep learning may provide higher accuracy in PD assessment than machine learning [42].

The TUG20 test accelerometer signals have a repetitive structure and contain gait features. Furthermore, there are two ways that the methods of AI could be applied. First, it can be used to collect a database of signals and split this database into two parts: the training and test sets. To increase the adequacy of the model, this approach might be applied after investigation of more than 100 PD cases, which is difficult in real life. The second way is to investigate the gait features and to understand what features are significant, i.e., to exclude insignificant features and thus decrease the dimension of the model, and then apply these data to clustering. This approach requires less studied cases, and we would prefer to follow it in the future. The major limitation of this study was the insufficient number of study subjects and measurements, which did not allow for a more precise analysis of data to be conducted. Furthermore, control groups (young and older healthy subjects) were not formed. In future studies, we propose that more measurements should be conducted in control groups and subjects with PD, both under “dry” immersion conditions and non-DI conditions.

## 5. Conclusions

In conclusion, the data from smartphone-based IMU accelerometers allowed us to compute gait characteristics that are conventionally used in the field of locomotion physiology, such as step duration and cadence. Like other IMU-based analyzing systems, the presented method allowed for the recognition of the phases of the TUG test. The application of an extended version of the TUG test provided a sufficient number of steps to characterize gait, and it allowed for the visualization of the duration of individual steps during the process of locomotion. Furthermore, the presented method appears to be suitable for a fast visual evaluation of stepping patterns in PD subjects. Of note, some of the specific characteristics of Parkinsonism events were recognized with the IMU sensors—for example, unusually long steps, which were produced while walking. 

For the entire group, the conditions of a single 45 min “dry” immersion affected none of the studied gait parameters derived with the help of smartphone-based IMU sensors; however, in one subject there was a clear increase in cadence, gait and turning speed. After a course of repeated DI sessions, some characteristics of the TUG test were improved; however, gait speed did not significantly change. 

The presented method of gait analysis appears to be suitable for further instrumentation because a smartphone is perfectly suited for association in IoT-based networks.

## Figures and Tables

**Figure 1 sensors-22-07915-f001:**
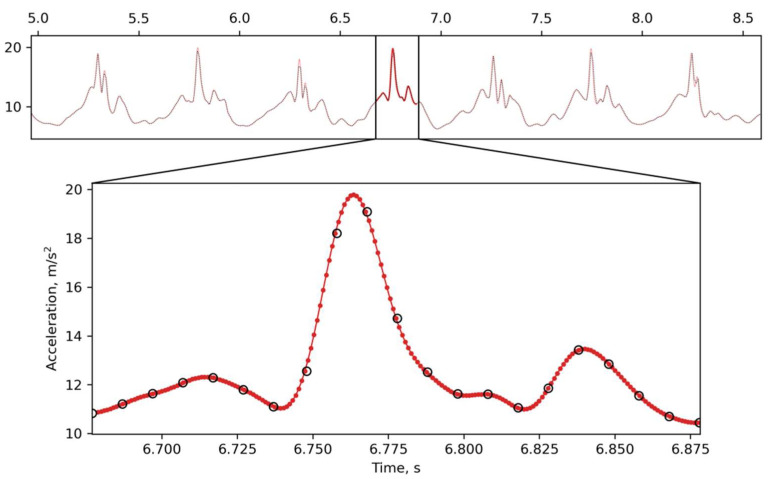
Up-sampling of a periodic signal obtained during walking. The black open circles on the bottom panel correspond to the real data sampled at a frequency of 100 Hz. The red points are new data points reconstructed with up-sampling. The red curve is a continuous signal passing through all the circles.

**Figure 2 sensors-22-07915-f002:**
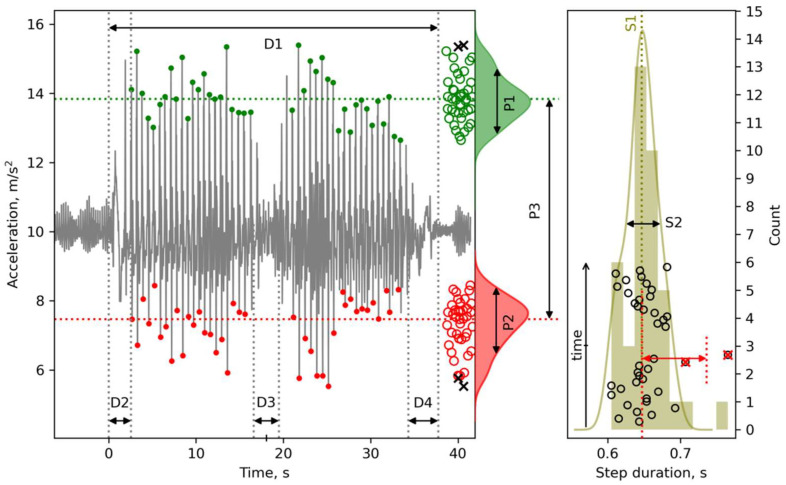
A representative plot of the 13 m TUG test with the phases (D1–D4) and parameters P1–P3 determined with an accelerometer (**left panel**) and the moments of step duration distribution (S1–S2, **right panel**) along progression of time.

**Figure 3 sensors-22-07915-f003:**
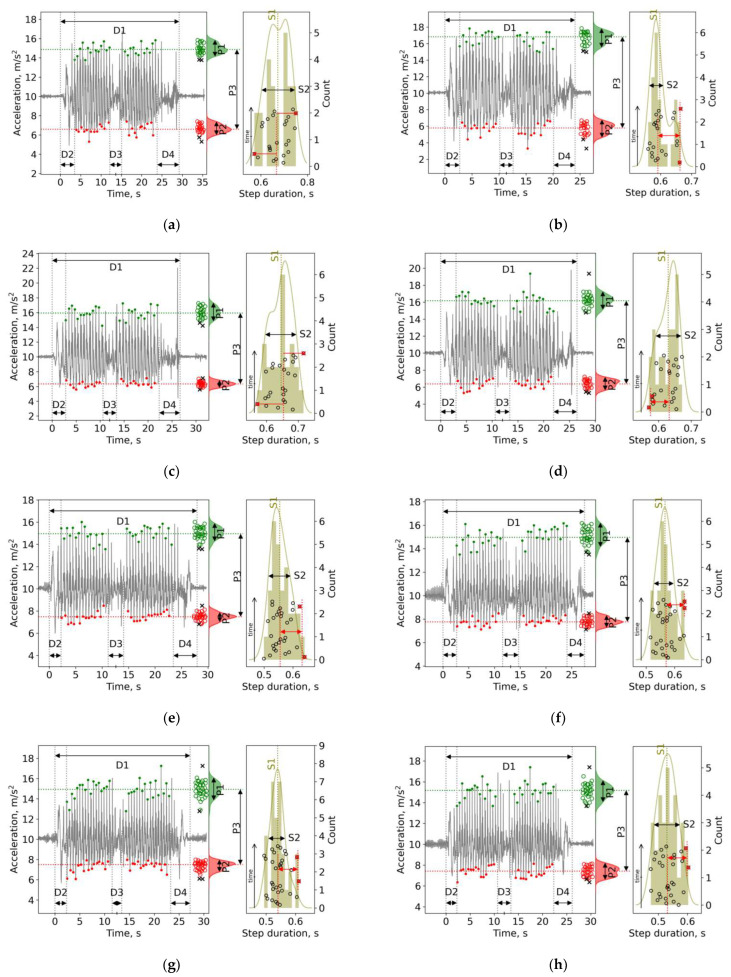
Individual plots for five separate DI sessions in three subjects. For details, see Figure 2: (**a**) Subject 1 before and (**b**) after the 1st DI; (**c**) Subject 1 before and (**d**) after the 7th DI; (**e**) Subject 2 before and (**f**) after the 5th DI; (**g**) Subject 2 before and (**h**) after the 4th DI; (**i**) Subject 3 before and (**j**) after the 7th DI.

**Figure 4 sensors-22-07915-f004:**
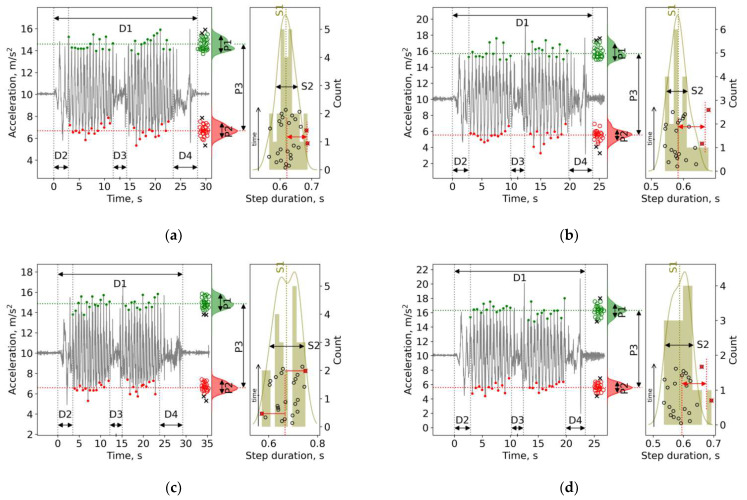
Individual plots for six courses of DI sessions in five subjects. For details, see Figure 2: (**a**) Subject 1 before and (**b**) after the 1st course of DI; (**c**) Subject 1 before and (**d**) after the 2nd course of DI; (**e**) Subject 2 before and (**f**) after the course of DI; (**g**) Subject 3 before and (**h**) after the course of DI; (**i**) Subject 4 before and (**j**) after the course of DI; (**k**) Subject 5 before and (**l**) after the course of DI.

**Table 1 sensors-22-07915-t001:** The anthropometric and clinical data of the subjects with PD.

No. of Subject	Age (year) and Gender	Height (cm), Weight (kg)	Stage by Hoehn & Yahr	Clinical Form
1	47 M	182, 81	1	PD, AR
2	61 M	188, 83	1	PD, T
3	58 F	158, 65	2	PD, T
4	50 M	171, 94	2	PD, T
5	55 M	178, 86	2	PD, T
6	69 M	170, 95	2	PD, T

T—tremulous form, AR—akinetic-rigid form of PD. Subject 1 participated in 2 courses of DI sessions. Subject 6 did not participate in DI sessions.

**Table 2 sensors-22-07915-t002:** Gait characteristics before and after a single DI session and after a course of DI sessions.

Parameter	Before a Single DI Session	After a Single DI Session	*p*	Before a Course of DI Sessions	After a Course of DI Sessions	*p*
D1 (s)	26.4 ± 3.2	25.3 ± 2.1	0.35	26.5 ± 2.3	24.7 ± 2.6	0.173
D2 (s)	2.56 ± 0.61	2.50 ± 0.44	0.89	2.66 ± 0.56	2.44 ± 0.35	0.116
D3 (s)	2.36 ± 0.52	2.60 ± 0.23	0.50	2.44 ± 0.45	2.27 ± 0.39	0.249
D4 (s)	4.19 ± 0.92	4.69 ± 1.34	0.69	4.09 ± 0.89	3.51 ± 0.64	0.046
S1 (s)	0.56 ± 0.10	0.55 ± 0.08	0.35	0.54 ± 0.08	0.52 ± 0.07	0.688
S2 (s)	0.027 ± 0.013	0.025 ± 0.01	0.89	0.02 ± 0.014	0.018 ± 0.01	1.000
S3 (steps/min)	109.1 ± 22.6	111.2 ± 18.3	0.69	112.6 ± 18.0	116.4 ± 16.6	0.249
S4 (steps/min)	4.19 ± 0.92	4.69 ± 1.34	0.50	3.64 ± 1.39	3.76 ± 1.63	0.753
S5	2.65 ± 0.54	3.37± 0.27	0.23	2.66 ± 0.61	3.26 ± 0.39	0.028
P1 (m/s^2^)	0.88 ± 0.40	0.95 ± 0.32	0.23	0.96 ± 0.40	0.92 ± 0.34	0.917
P2 (m/s^2^)	0.46 ± 0.08	0.56 ± 0.18	0.23	0.46 ± 0.11	0.61 ± 0.19	0.075
P3 (m/s^2^)	9.2 ± 2.26	9.9 ± 2.8	0.35	9.7 ± 2.35	10.75 ± 1.91	0.116

*p*—probability of difference in Wilcoxon test between “before” and “after” conditions. The meaning of D-, S- and P-parameters can be found in the text.

## Data Availability

The datasets generated for this study are available on request to the corresponding author.

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
