# Peer review of "Gait Characteristics Analyzed with Smartphone IMU Sensors in Subjects with Parkinsonism under the Conditions of “Dry” Immersion"

_sensors, 2022, doi:10.3390/s22207915_

Round 1

Reviewer 1 Report

From the technological point of view there are serious problems with the drifts mainly in the gyroscope signals and accuracy due in the accelerometer signals. No action is presented to compensate for this fact.

It’s written: “Five subjects with PD participated in the study on their informed consent”. But you use 6 patients in your study. Could you clarify?

No description neither picture or scheme related to the exact position of the smartphone on the head. No analysis of the comfortability or acceptance of such position

No explanation about how to synchronize the gyroscope and accelerometer of the smartphone 

Turn detection based on empirical threshold with a very small database

The measurement of the duration of the TUG phases needs the allocation of the critical points like for example, the starting of the sit to stand movement, the determination of the beginning of the first step, among others. There is no explanation neither algorithm description of the detection of this important points.

The detection of two outliers for the step length variable is justified mathematically but not the causes that cause them

It’s written: “The two longest steps were excluded from averaging statistics. However, they probably 301 informed on light form of "freeze of gait". Freezing of gait (FOG) provokes very short steps and tremor in the legs, completely contrary to what is stated. Additionally, FOG is normally present at the beginning of movement, prior to a turn or at the end of a turn, areas that are discarded for the analysis of the evolution of the step length. This is a contradiction.

Author Response

Dear Reviewer,

please find below our answers to your comments:

From the technological point of view there are serious problems with the drifts mainly in the gyroscope signals and accuracy due in the accelerometer signals. No action is presented to compensate for this fact.

Action taken:

New paragraph was added to the text of 2.4. Section (right below Figure 1):

"A simple calibration of the zero offsets of the sensors was performed before the start of the test. To do this, we used the measurements obtained from a smartphone placed on a horizontal surface. It was noted that the sensor offsets was probably pre-calibrated by the Android OS. Bias instability and velocity/angle random walk for smartphone sensors was previously analyzed by us using the Allan variation [14]. Bias instabilities are (7.3, 8.2, 8)*10-4 m/s2 for x/y/z axes of accelerometer, (1.7, 5, 7) *10-5 deg/s for gyroscope. Since the test duration is less than 1 minute, bias drifts can be neglected".

  1. It’s written: “Five subjects with PD participated in the study on their informed consent”. But you use 6 patients in your study. Could you clarify?

Answer:

That's right, only five subjects participated in the study with "dry" immersion intervention. Subject No.6 also was going to participate, he signed his informed consent, participated in preliminary measurements, including TUG test, but then stopped his communication with us (actually, was unavailable). The pattern of his walking was very peculiar (steps were variable, and accelerometric metrics were changing along the Go-phases). We decided to leave the data on this subject in the study for two reasons: 1) to present the method (figure 2, which presents the accelerometric data, is drawn using the data from subject 6); 2) to better characterize gait characteristics with the presented method in PD (6 subjects constitute a bit bigger group than 5 subjects ), regardless of "dry" immersion.

Action taken:

"Five subjects" is substituted by "Six subjects" in the second sentence of the Subjects subsection of Introduction.

  1. No description neither picture or scheme related to the exact position of the smartphone on the head. No analysis of the comfortability or acceptance of such position.

Answer:

The smartphone was fixed on the back of the head of a subject with an elastic band and, additionally, tight knitted hat. Subjects felt comfortable with this kind of fixation. Smartphone has never fallen out of its position.

Action taken:

We added the sentences "The smartphone was fixed on the back of the head of a subject with an elastic band and, additionally, tight knitted hat. Subjects felt comfortable with this kind of fixation. Smartphone has never fallen out of its position.

  1. No explanation about how to synchronize the gyroscope and accelerometer of the smartphone

Action taken:

We modified a paragraph to the text of 2.4. Section (Data processing):

"Thus, the measurements were accompanied by timestamps from the smartphone's operating system timer. For further analysis, the accelerometer and gyroscope measurements with a time-stamp difference of less than 5 ms (half of measurement period) were considered synchronous".

  1. Turn detection based on empirical threshold with a very small database

Answer:

In this particular study, we primarily aimed at assessment of gait characteristics. Stepping was clearly seen by accelerometer signal, which allowed identifying both Gait-Go and Gait-Come phases. However, turn detection is also important, because turning in PD subjects is believed to be of even higher diagnostic value than straightforward stepping (Hasegawa et al., 2019). In that respect, we have to agree that the database is small for detection of turns. We suppose to expand our data base and focus on this issue in future studies.

  1. The measurement of the duration of the TUG phases needs the allocation of the critical points like for example, the starting of the sit to stand movement, the determination of the beginning of the first step, among others. There is no explanation neither algorithm description of the detection of this important points.

Action taken:

A paragraph was added to the text of 2.3. Section (data processing): "The subject was instructed to sit still and look forward before and after the test. At motionless state, the shape of the accelerometer signal is formed by the current projections of the gravity vector, measurement noise and the existing zero-G offsets, as well as head tremor. The beginning and end of the movement are characterized by a change in the x-, y-, z- components of the acceleration vector due to the inclination as well as the presence of linear accelerations at standing up and sitting down. For the gyroscope at motionless state, the signal includes sensor noise, as well as head tremor. However, it is characterized by the constancy of the mean value (measurement offset). The start and end points of the TUG test were selected manually analyzing the change in the mean value due to rotation of head and body during inclination at standing up and sitting down".

  1. The detection of two outliers for the step length variable is justified mathematically but not the causes that cause them.

Answer:

These two outliers (two longest steps) were identified for the reason that they were the characteristic of "transitional" moments of TUG test, for example, first steps at the beginning of Gait-Go phase (figures 3 and 4), two steps right before the U-turn (see Fig. 2), or by the end of the Gait-Come phase (figures 3-4). These outliers were reporting on the first, second, last before U-turn, last, the second last step. They were not informing on MEAN cadence during Gait pahases. Thus, we decided to discard them from the analysis of gait.

Action taken:

A sentence is added to the 2.7.1. section (S2 parameter): "Usually, these outliers were the characteristic of the "transitional" moments of TUG test (at the beginning and end of Gait-Go and Gait-Come phases, and before the U-turn)

  1. It’s written: “The two longest steps were excluded from averaging statistics. However, they probably informed on light form of "freeze of gait". Freezing of gait (FOG) provokes very short steps and tremor in the legs, completely contrary to what is stated. Additionally, FOG is normally present at the beginning of movement, prior to a turn or at the end of a turn, areas that are discarded for the analysis of the evolution of the step length. This is a contradiction.

Answer:

We agree with that comment. Of course, FOG means exactly what the reviewer wrote (short steps and, eventually, stop of walking). The phenomenon (symptom) "freezing of gait" (FOG) was used (incorrectly) in earlier versions of the manuscript and further all mentions of FOG were deleted, but that mistake was missed.

Action taken:

Sentence "However, they probably informed on light form of "freeze of gait" is deleted from the text. In addition, the acronym FOG was deleted from the section 2.7 (Gait characteristics) and substituted by LS (unusually Long Steps). In general, this parameters (S5, or LS) informs that stepping is not rhythmic, with unpredictable lengths (duration) of steps.

In addition, 9 new references were added to the text.

Reviewer 2 Report

The paper presents a study to evaluate gait characteristics by analyzing the accelerometer signal received from a smartphone attached to the head during an extended TUG test for Parkinson's disease patients.

The topic is interesting. The paper is well-organized, but, the presentation quality, especially for images, could be improved. However, I think that the paper cannot be accepted because it is out of the scope of the special issue. Indeed, the paper reports the description of a study without discussing the application of neural network-based approaches or, artificial intelligence techniques, in general. Hence, it is difficult to understand the contribution of the paper.

Author Response

Dear Reviewer,

please find below our answers to your comments:

The paper presents a study to evaluate gait characteristics by analyzing the accelerometer signal received from a smartphone attached to the head during an extended TUG test for Parkinson's disease patients. The topic is interesting. The paper is well-organized, but, the presentation quality, especially for images, could be improved.

Answer and action taken:

The quality of images is improved.

However, I think that the paper cannot be accepted because it is out of the scope of the special issue. Indeed, the paper reports the description of a study without discussing the application of neural network-based approaches or, artificial intelligence techniques, in general. Hence, it is difficult to understand the contribution of the paper.

Answer:

Yes, this study contributes entirely to data collection and signal analysis of gait  in PD. However, we regarded that this paper would be of interest to reader because smartphones are extremely suitable for application in neural network-based approaches, as smartphones are already a part of the Internet. This paper was a first step in this direction.

Action taken:

In the Discussion section, we added four paragraphs of consideration on that issue.

In addition, 9 new references were added to the text.

Reviewer 3 Report

Overall this is a well written manuscript. Below are my comments

Introduction

The introduction is well written and does a great job of introducing the literature. My only comment is that there are some very minor grammatical errors (i.e. missing "the" or "a/an")

Methodology

The methodology needs additional information

1. Did you just go back and recruit fro your previous study? Or did you just use the data of these 6 subjects?

2. You said that 5 subjects with PD participated in the study on their informed consent. I don't quite understand that statement. What about subject number 6? Why not just remove subject 6 completely from the study since they did not participate in DI? You can state we recruited 6, but only 5 participated in the study.

3. How longis several minutes? Please specify. Perhaps rephrasing it to state "subjects laid supine until BP was < 140/80"??

4. Perhaps I am missing this part of the experiment, but did you just do a pre-post study? Or are you only interested in post-DI gait data? In your procedures section please provide a section that specifically lays out the study protocol (i.e. baseline gait, daily gait analyses post-DI, etc)

5. How long after 1 session did you measure gait?

6. Line 192, please use the name of the complementary filter

7. In the statistical analyses did you correct for multiple analyses?

Results

The results are well presented. The resolution of the figures is not great. I would suggest working with the editors to improve them (I have had this happen in Sensors before, so please work with the editors to make sure that when they're getting converted to PDF the resolution is not reduced)

Discussion

1. I don't understand how this study measured reliability. You did not compare the results to other methods so you did not measure reliability

2. I think when you talk about the inter-individual differences, it might be better to present the inter-individual data in your tables. I know the figures do that, but the figures are harder to read.

3. I think it would make more sense to present individual differences rather than trying to use statistics to determine differences as you stated you had a lot of inter-subject variability with 1 out of your 5 subjects not adhering to part of your protocol. 

Additional thoughts:

I would highly recommend examining the sit to stand (velocity as well as trunk movement) aspect of the modified TUG. Our work has shown significant changes with various interventions (the n's for our studies usually range between 11 and 22).

Author Response

Dear Reviewer,

please find below our answers to your comments:

Introduction

The introduction is well written and does a great job of introducing the literature. My only comment is that there are some very minor grammatical errors (i.e. missing "the" or "a/an")

Methodology

The methodology needs additional information

  1. Did you just go back and recruit for your previous study? Or did you just use the data of these 6 subjects?

Answer:

In this manuscript, we just present measurements, which were conducted along with all the other measurements (heart rate variability, EMG, posturography, reaction time, clinimetry, etc). So, the results are taken from same cohort of Parkinson's disease patients (altogether 26 subjects). In five of them, a smartphone was used to evaluate gait.

Action taken:

To clarify this issue, we added a sentence "The data on gait characteristics presented in this article were obtained from these studies" right after "All of them are from the same cohort of subjects who participated in our earlier studies [8-10, 23]".

  1. You said that 5 subjects with PD participated in the study on their informed consent. I don't quite understand that statement. What about subject number 6? Why not just remove subject 6 completely from the study since they did not participate in DI? You can state we recruited 6, but only 5 participated in the study.

Answer:

Yes, only five subjects participated in the study with "dry" immersion intervention. Subject 6 also was going to participate, he signed his informed consent, participated in preliminary measurements, including TUG test, but then stopped his communication with us (actually, has withdrawn from the study). Still, the pattern of his walking was very peculiar (steps were variable, and accelerometric metrics were changing along the Go-phases). We decided to leave this subject in the study for two reasons: 1) to present the method (figure 2, which presents the accelerometric data and analysis is drawn using the data from subject 6); 2) to better characterize gait characteristics with the presented method in PD (6 subjects constitute a bit bigger group than 5 subjects ), regardless of "dry" immersion.

Action taken:

"Five subjects" is substituted by "Six subjects" in the second sentence of the Subjects subsection of Introduction.

  1. How long is several minutes? Please specify. Perhaps rephrasing it to state "subjects laid supine until BP was < 140/80"??

Answer

Altogether, it took around 10 minutes from the moment of lying supine to the start of the "dry" immersion session. Around 5 minutes were used for attaching ECG electrodes, BP measuring and familiarization, another 5 minutes (strictly) were reserved for ECG monitoring/recording. Most of subjects had the arterial hypertension (AH) and, therefore, were taking anti-AH therapy. This allowed them to achieve the control values of BP (less than 140/90 mm Hg). In the supine position on a horizontal surface, BP is lower than during vertical stance. If BP exceeded the control values by the end of 10 minutes lying supine, we did not wait further for its decrease. We just banned the session, and postponed it for another day.

Action taken:

Two corresponding sentences are re-written: "Before immersion, subjects laid supine for approximately 10 minutes on a solid movable motor-driven platform of the MEDSIM facility on a cotton sheet for attaching electrocardiogram electrodes, measuring brachial blood pressure (BP), and familiarization (altogether around 5 min), and ECG recording in standard lead II (5 min). If by the end of 10 minutes lying supine the subject's BP was higher than 140/80 mm Hg, he/she was not allowed to enter DI, and the study was postponed for another day".

  1. Perhaps I am missing this part of the experiment, but did you just do a pre-post study? Or are you only interested in post-DI gait data? In your procedures section please provide a section that specifically lays out the study protocol (i.e. baseline gait, daily gait analyses post-DI, etc)

Answer:

Yes, the study is designed as a pre-post one (pre-DI and post-DI study points were compared). We were interested namely in post-DI data. As of now, we see that collecting more data (on daily, basic level) on gait would be interesting and helpful, but, unfortunately, we did not do that.

Action taken:

To address this comment, we added a new sentence to the 2.2.2. subsection (TUG test): " Baseline gait analysis throughout the day was not conducted, neither before nor after the DI session.

  1. How long after 1 session did you measure gait?

Answer:

We measured gait up to 10 minutes after the DI session, including 5-7 minutes of re-adapting to usual conditions (lying on a platform right after DI). Other 2-3 minutes were used to attach a smartphone on the head and to instruct a subject (ones again) how to proceed with the TUG test. The room where DI was conducted was open to the corridor, where the TUG test was performed, which means very close.

Action taken:

A sentence was added to the 2.2.2. section: "The TUG test was performed prior and some 8-10 min after the DI session".

  1. Line 192, please use the name of the complementary filter

Action taken:

"proposed by Robert Mahony et al." is added to the sentence "The orientation of the smartphone was calculated using a well-known complementary filter proposed by Robert Mahony et al. [26] and is expressed in the form of a quaternion Q.

  1. In the statistical analyses did you correct for multiple analyses?

Answer: No, we did not use corrections as the analyses were not multiply (paired Wilcoxon test).

Results

The results are well presented. The resolution of the figures is not great. I would suggest working with the editors to improve them (I have had this happen in Sensors before, so please work with the editors to make sure that when they're getting converted to PDF the resolution is not reduced)

Answer: We improved the resolution of figures.

Discussion

  1. I don't understand how this study measured reliability. You did not compare the results to other methods so you did not measure reliability

Answer:

We added a brief review of current methods to study motion in PD.

  1. I think when you talk about the inter-individual differences, it might be better to present the inter-individual data in your tables. I know the figures do that, but the figures are harder to read.

Answer:

We decided that adding a Supplement Material (S1) with all individual data would be helpful to read the result.

Action taken:

Supplement 1 (Table with individual data) is added to the manuscript.

  1. I think it would make more sense to present individual differences rather than trying to use statistics to determine differences as you stated you had a lot of inter-subject variability with 1 out of your 5 subjects not adhering to part of your protocol.

Answer: Yes, inter-subject variability is high.

In the Results section we paid attention to such individual peculiarities:

"For example, in Subject 1, after a single one DI, the D1 (duration of the entire TUG test) has decreased by 5 sec - from 30 to 25 sec, his cadence increased from 88 to 100 steps/min. After another DI session in this subject, the values of P1 parameter (heel strike) increased from 0,55 to 0,77 m/sec2, P2 (swing phase) - from 0,50 to 0,82 m/sec2, and P3 - from 8-9 to 10-11 m/sec2 (Fig. 3), which informs on increased variability of step length after DI and stronger strike of the heel on the floor. In Subject 2, only the value of D4, and in Subject 3 - the value of S5 has decreased (Fig. 3). In addition, in all 5 cases of DI, the change of the distribution type of P2 from the unimodal distribution to a more bimodal one took place (see Fig. 3)".

Additional thoughts:

I would highly recommend examining the sit to stand (velocity as well as trunk movement) aspect of the modified TUG. Our work has shown significant changes with various interventions (the n's for our studies usually range between 11 and 22).

Answer: We fully agree with that comment. Earlier, we have evaluated duration of the sit-to-stand phase after the "dry" immersion session with motion videocapture (Meigal, A., Tretjakova, O., Gerasimova-Meigal, L., Prokhorov, K., & Sayenko, I. (2021). Motion videocapture and treadmill to study postural reactivity and transition: Application to the condition of 'dry' immersion in Parkinson's disease. Conference of Open Innovation Association, FRUCT, 2021-October 141-147. doi:10.23919/FRUCT53335.2021.9599994

We did not find difference between pre- and post-DI study points. Still, it would make sense to further analyze this phase also with accelerometers. The data can be extracxted from the present study.

In addition, 9 new references were added to the text.

Round 2

Reviewer 1 Report

I accept the explanation and correction done in the article

Reviewer 3 Report

I appreciate the authors taking all my suggestions into account. This manuscript is significantly better. I also appreciate the authors posting their supplementary files.